# Resistance of Modern Russian Winter Wheat Cultivars to Yellow Rust

**DOI:** 10.3390/plants12193471

**Published:** 2023-10-03

**Authors:** Elena Gultyaeva, Ekaterina Shaydayuk

**Affiliations:** All Russian Institute of Plant Protection, Shosse Podbelskogo 3, St. Petersburg 1986608, Russia; eshaydayuk@bk.ru

**Keywords:** *Puccinia striiformis*, resistance, *Triticum aestivum*, *Yr* genes

## Abstract

Over the last decade, the significance of yellow rust caused by *Puccinia striiformis* (*Pst*) has substantially increased worldwide, including in Russia. The development and cultivation of resistant genotypes is the most efficient control method. The present study was conducted to explore the yellow rust resistance potential of modern common winter wheat cultivars included in the Russian Register of Breeding Achievements in 2019–2022 using the seedling tests with an array of *Pst* races and molecular markers linked with *Yr* resistance genes. Seventy-two winter wheat cultivars were inoculated with five *Pst* isolates differing in virulence and origin. Molecular markers were used to identify genes *Yr2*, *Yr5*, *Yr7*, *Yr9*, *Yr10*, *Yr15*, *Yr17*, *Yr18*, *Yr24*, *Yr25* and *Yr60*. Thirteen cultivars were resistant to all *Pst* isolates. The genes *Yr5*, *Yr10*, *Yr15* and *Yr24* that are effective against all Russian *Pst* races in resistant cultivars were not found. Using molecular methods, gene *Yr9* located in translocation 1BL.1RS was detected in 12 cultivars, gene *Yr18* in 24, gene *Yr17* in 3 and 1AL.1RS translocation with unknown *Yr* gene in 2. While these genes have lost effectiveness individually, they can still enhance genetic diversity and overall yellow rust resistance, whether used in combination with each other or alongside other *Yr* genes.

## 1. Introduction

Wheat is one of the most important crops worldwide, and it grows in various climatic zones [1]. Common (or bread) wheat (*Triticum aestivum*) is the one most dominant among cultivated *Triticum* spp. About 95% of wheat produced worldwide is common wheat [2], and in Russia it represents 36% of all grain crops produced [3]. In Russia, winter wheat predominates, and it is grown in the North Caucasus, Central Chernozem and central regions, whereas spring wheat is grown in the Volga region, western Siberia and the Urals [4]. The areas of winter wheat cultivation are characterized by high soil and climate diversity; in this regard, the requirements for wheat cultivars in each area are highly specific. In modern wheat breeding, special attention, along with the improvement of the main economically valuable traits, is given to increasing resistance to harmful organisms (pathogens and pests). Growing resistant cultivars is the most efficient and environmentally friendly method to reduce yield losses. The availability of commercial cultivars with various types of resistance and the corresponding controlling genes provides effective protection. One significant constraint for increasing wheat production is the rust pathogens, leaf rust (*Puccinia triticina*), stem rust (*Puccinia graminis*) and yellow (stripe) rust (*Puccinia striiformis* (*Pst*) [5]. Long-distance dispersal capacity, rapid development of virulence and climate adaptability make wheat rusts the most important threat to wheat production worldwide [6]. If any of these rusts reach epidemic proportions, devastating yield losses can occur, with total loss potentially occurring in fields with highly susceptible cultivars [7]. *P. triticina* is the most distributive rust species and is able to develop over a wide temperature range. Up until 2000, leaf rust caused serious epidemics throughout wheat-growing regions worldwide. Stem rust pathogens can also occur wherever wheat is grown [8], but it is a more thermophilic species. Maximum temperature for spore germination and sporulation is about 5.5 °C higher than for *P. triticina.* Stem rust differs from leaf rust in requiring a longer dew period [8].

The environmental requirements of *Pst* differ from *P. triticina* and *P. graminis*. Yellow rust is generally found in northern latitudes, highlands and wheat-growing regions with cooler temperatures during early growth stages. However, recent large-scale epidemics have occurred in warmer wheat-growing areas. This was ensured by the emergence of two closely related *Pst* strains with increased aggressiveness and tolerance to warm temperatures [7,9,10].

In 2000, repeated *Pst* epidemics occurred in the majority of wheat-growing areas in Western Europe [9,10,11,12], Central and East Asia [13,14], the Middle East, North and South Africa [13], North and South America [15] and Australia [16]. Notably, the rapid global emergence of more aggressive and genetically diverse *Pst* populations adapted to warmer temperatures had an impact on the yellow rust resistance ratings of many wheat cultivars [9]. Therefore, widening the genetic diversity of wheat cultivars in current use would be viewed as a strategy for greater yield stability.

Yellow rust epidemics in Russia were for the most part only frequent and destructive in the North Caucasus [17]. Over the last 5 years, the significance of *Pst* has increased markedly in other Russian regions. The disease has appeared in the regions North-West [18], West Siberia [19], Volga [20] and Central Chernozem regions [21]. Global climate changes contribute to changes in disease incidence and severity. There are many ways to manage rust diseases; however, the development and cultivation of resistant genotypes are the most efficient [7].

Resistance to rust in wheat is grouped into two broad categories. Seedling or all-stage resistance is often differentially expressed and is commonly designated as race-specific resistance. Race-specific resistance with a single resistance gene is often short-lived. The second type of resistance, often designated as adult plant resistance (APR), is effective at post-seedling or adult plant stages and is usually manifested by a slow disease progression (slow-rusting) despite a susceptible host reaction. However, some race-specific adult plant resistance genes have also been identified. Effective levels of slow-rusting resistance are more commonly controlled by a small number of minor genes with additive effects, and some have pleiotropic effects in conferring resistance to other diseases [22,23].

Presently, 84 yellow rust resistance genes (*Yr* genes) have been catalogued in wheat, with these mostly conferring to all-stage (or seedling) resistance [24]. This kind of resistance generally provides qualitative resistance to one or more *Pst* isolates [22]. Some of the catalogued seedling resistance *Yr* genes are alien and obtained from diploid and tetraploid wild and cultivated relatives: *Yr15*, *Yr35* and *Yr84* derived from *Triticum dicoccoides*, *Yr8* from *Aegilops comosa*, *Yr9* from *Secale cereale*, *Yr10* from *T. vavilovii*, *Yr17*, *Yr28* and *Yr48* from *Ae. ventricosa*, *Yr37* from *Ae. kotschyi*, *Yr38* from *Ae. sharonensis*, *Yr40* from *Ae. geniculata*, *Yr42* from *Ae. neglecta*, *Yr50* from *Thinopyrum intermedium*, *Yr69* from *Th. Ponticum, Yr70* from *Ae. umbellulate* and *Yr5* from *T. spelta* [24,25,26]. Other genes came from hexaploid wheat landraces. Of the race-specific genes, *Yr5*, *Yr10*, *Yr15* and *Yr24* are still widely effective and can be used in breeding for yellow rust resistance [13,27]. APR confers quantitative or partial resistance. It is not typically race-specific, is consequently more durable than seedling resistance and is conditioned by genes with minor or partial effect. Some APR genes confer resistance against multiple biotrophic pathogens; this characteristic is a good indicator of durability. Examples include *Yr18/Lr34/Sr67/Pm38*, *Yr29/Lr46/Sr58/Pm39*, *Yr30/Lr27/Sr2* and *Yr46/Lr67/Sr55/Pm46* [28]. Pyramiding multiple resistance genes with additive effects into single genetic backgrounds should help prevent a rapid breakdown of wheat *Pst* field resistance [29].

In most countries, cultivars have been assessed for disease resistance for registration to the National Catalogue [22], but seedling resistance and characterization of the *Yr* genes are not performed. The field resistance of Russian wheat genotypes to more important diseases is also evaluated during preliminary state variety studies. The complementary monitoring of wheat at the seedling stage for resistance to *Pst* pathotypes has been essential for the detection of race-specific resistance. Assessment based on tests using a set of *Pst* isolates differenced for virulence and molecular markers used for the postulation of *Yr* genes’ molecular marker-assisted detection is the most convenient and reliable method to identify the presence of *Yr* genes. A wide range of markers are reported to be associated with *Yr* genes in wheat [30]. Distribution of *Yr* genes in commercial wheat cultivars and breeding lines using molecular markers is characterized in many countries [22,31,32,33]. Information about the postulation of *Yr* genes in Russian commercial cultivars using multi-pathogen tests and molecular markers is too limited.

The present study was conducted to explore the yellow rust resistance potential of modern common winter wheat cultivars included in the Russian Register of Breeding Achievements using the seedling tests with an array of *Pst* races and molecular markers linked with *Yr* resistance genes.

## 2. Results

### 2.1. Resistance Study at the Seedling Test

Virulence and avirulence spectra of *Pst* isolates on Avocet near-isogenic lines (NILs) and a set of differential cultivars are presented in Table 1. All isolates were avirulent to Avocet lines (Av*Yr*) with genes *Yr5*, *Yr10*, *Yr15*, *Yr24*, *Yr26* and *YrSp* and differential cultivars (cvs Moro, Spaldings Prolific and Nord Desprez). Avocet NILs with genes *Yr8*, *Yr9* and *Yr18* and cvs Lee, Heines Kolben, Vilmorin 23, Suwon 92/Omar, Jupateco S and Avocet S were susceptible. *Pst* isolates differed in their avirulence to *Yr1*, *Yr7*, *Yr17* and *Yr27*, and cvs Chinese 166 (*Yr1*), Hybrid 46 (*Yr4*), Reichersberg 42 (*Yr7*) and Heines Peko (*Yr2*, *Yr6*, *Yr25*), Compair (*Yr8* and *Yr19*) and Carstens V (*Yr32*, *Yr25*). Therefore, these genes could be used for gene postulation in multi-pathogen tests. Avirulence to *Yr1* and cv. Chinese 166 (*Yr1*) was detected for North-Western *Pst* isolate; to *Yr7* for North-Western, West Siberian and Derbent *Pst* isolates; to *Yr7* for West Siberian, Krasnodar and Rostov *Pst* isolates; to *Yr27* for Krasnodar and Rostov *Pst* isolates; to cv. Strubes Dickkopf for North-Western and Derbent *Pst* isolates; to cv. Hybrid 46 for North-Western and West Siberian isolates; to Reichersberg 42 for West Siberian *Pst* isolate; to Heines Peko for Derbent *Pst* isolate; to Compair for all North Caucasian *Pst* isolates (Derbent, Krasnodar and Rostov); and to Carstens V for North-Western and West Siberian *Pst* isolates.

Seedling infection type data for the 72 tested winter wheat cultivars inoculated with five *Pst* isolates are listed in Table 2. The resistance of the cultivars varied greatly when infected with various isolates (infection type (IT) from 0 to 2). Thirteen cultivars (cvs Timiryazevka 150, Akapella, Vol’nitsa, Yelanchik, Galateya, Polina, Khamdan, Sharm, Yubiley Dona, Pal’mira 18, Podruga, Sirena and Fyodor) were resistant to all *Pst* isolates tested and 30 were highly (IT 3–4) or moderately susceptible (IT 2+–3 and 3). The number of winter wheat cultivars resistant to the West Siberian *Pst* isolate was significantly higher (46%) than to North Caucasian and North-Western *Pst* isolates. North Caucasian *Pst* isolates used in the multi-pathogen tests were more virulent (virulence to 17 and 18 differentials from 30) than those from West Siberia and North-West (15).

According to McIntosh et al. [5], highly elevated levels of seedling resistance are almost invariably associated with high or acceptable levels of adult plant resistance under field conditions. Intermediate seedling responses may give variable responses in the field. This is consistent with the results obtained in these studies. Most of cultivars resistant to yellow rust at the seedling stage were resistant at the adult plant stage under field conditions in state-run cultivar tests [34,35,36,37,38] (Table 2).

A resistant reaction to only one of the used *Pst* isolates occurred in 15 cultivars. Cvs Arsenal, Al’bireo and Leo were resistant to the North-Western isolate, which differed from other isolates by avirulence to *Yr1*. According to multi-pathogen tests, it is possible to postulate the presence of the *Yr1* gene in these cultivars. However, infection types of these cultivars under inoculation of North-Western *Pst* isolate was moderately resistant (IT 1–2; and 2;) in comparing to cv. Chinese 166 and Av*Yr1* (IT 0). Cvs Bodryy, Videya, Gerda, Stat’, Shef, Akhmat, Bogema, Partner, Rifey and Rossyp’ were resistant to a West Siberian isolate avirulent to cv. Reichersberg 42; cvs Etyud and En Mars to Derbent isolate avirulent to cv. Heines Peko (*Yr2*).

A resistant reaction to two *Pst* isolates characterized four cultivars. Cvs Bylina Dona, Volodya and Timiryazevskaya yubileynaya were resistant to Krasnodar and Rostov *Pst* isolates avirulent to *Yr27* and cv. Bumba was resistant to North-Western and West Siberian *Pst* isolates avirulent to cvs Hybrid 46 and Carstens V. A resistant reaction to three *Pst* isolates characterized seven cultivars. Cvs Iridas, Markiz, Gomer, Moskovskaya 82 and Nemchinovskaya 85 were resistant to West Siberian and North Caucasian *Pst* isolates from Krasnodar and Rostov avirulent to *Yr17*; cv. V’yuga was resistant to three North Caucasian *Pst* isolates avirulent to cv. Compair and cv. Status were resistant to North-Western, West Siberian and Derbent *Pst* isolates avirulent to *Yr7*. Cvs Anastasiya and En Foton were resistant to West Siberian and all North Caucasian isolates.

The reaction type of 13 Russian cultivars resistant to all used *Pst* isolates varied from 0 to 2. At the same time, the Avocet NILs with genes *Yr5*, *Yr10 Yr15* and *YrSp* and cv. Spalding Prolific were highly resistant (score 0) (Table 1). This indicates that the genes of the studied cultivars differ from the *Yr5*, *Yr10* and *Yr15* genes. Response to inoculation for Avocet NILs with genes *Yr24* and *Yr26* fluctuated from 0 to 2. Middle resistant types (2, 2;) were detected during inoculation by North-Western *Pst* isolate, resistant (0–1) by North Caucasian isolates and highly resistant (0) by West Siberian isolate. The type of reaction of the Russian resistant cultivars differed from Avocet NILs and supplemental wheat differentials, which indicates their differing genetic control of resistance.

### 2.2. Detecting Yellow Rust Resistance Genes Using Molecular Markers

Identification of genes *Yr5*, *Yr10*, *Yr15* and *Yr24* was conducted for twelve resistant cultivars (Timiryazevka 150, Akapella, Vol’nitsa, Yelanchik, Galateya, Polina, Sharm, Yubiley Dona, Pal’mira 18, Podruga, Sirena and Fyodor). These genes were highly effective to all Russian *Pst* isolates [39]. Markers STS7/8 and STS9/10 linked to *Yr5* and markers Xpsp3000, Xbarc8 and Barc181 linked to *Yr10*, *Yr15* and *Yr24*, respectively, were not amplified on the genomic DNA of any of the resistant cultivars indicating the absence of genes. The results of molecular studies are consistent with multi-pathogen tests, where was showed that the reaction type of these cultivars differed from Avocet NILs and differential cultivars with these genes.

Marker-assisted detection of genes *Yr2*, *Yr7*, *Yr9*, *Yr17*, *Yr18*, *Yr25* and *Yr60* was performed for all winter wheat cultivars. Some of these genes had lost their effectiveness for disease protection, but they could still play a role in multiple gene resistances in countries where the frequency of virulence is quite low or absent, or in combination with other *Yr* genes.

SSR marker Wmc364 was used for the identification of *Yr2.* Genetic distance among marker Wmc364 and resistance gene *Yr2* was calculated as 5.6 cM [40]. A different size of amplified alleles for this marker is present in the publication. According to Rani et al. [32], *Wmc364* was +200 bp/−190 bp for distinguishing gene positive and negative genotypes, but according to Feng et al. [40], it was −207 bp/+201 bp. In our study, the cv Heines Kolben had an amplicon of approximately 190 bp. Cvs Heines Peko, Heines VII and all Russian winter genotypes yielded a fragment of a low size (Appendix A). It is assumed that *Yr2* is absent in the wheat collection studied. However, the difference between positive and negative genotypes was too low. The reason for this may be the large distance between the marker and the gene.

Microsatellite marker CFD77 was used for the identification of *Yr7* [32]. According to McIntosh et al. [5], *Yr7* is allelic or closely linked with *Yr5.* In our *Pst* virulence surveys [39], the reaction of NILAv*Yr7* and cvs Lee and Reichersberg 42 having also *Yr7* gene differs strongly from NILAv with gene *Yr5.* Of the infection types used in this study, *Pst* isolates varied from moderately resistant to susceptible for NILAv*Yr7* and cv Reichersberg 42 but were highly susceptible (IT 3 and 4) for cvs Lee. The reaction of NIL Av*Yr5* was always highly resistant (IT 0), which indicates the differences between these genes. SSR marker CFD77 amplified a 220-bp allele in genotypes with *Yr7*. This fragment was detected in all Russian cultivars and differentials (NIL Av*Yr7*, Lee and Reichersberg 42). Similar results were reported by Rani et al. [32] from molecular studies of Indian wheat genotypes. However, in our study, this marker amplified an additional fragment in positive differentials with a size near 300 bp, but it was not detected in the wheat cultivars studied (Appendix A).

Marker SCM9 was selected to identify *Yr9*. The rye microsatellite marker SCM9 was able to detect different sources of wheat-rye translocations involving 1BL.1RS with *Yr9* gene and 1AL.1RS with unknown *Yr* gene. A band of 228 bp was associated with genotypes with 1AL.1RS, and a 207-bp band in those with *Yr9* (Appendix A) [41]. Twelve wheat cultivars (17%) had amplicon 228 bp indicating the presence of gene *Yr9,* and two cultivars had amplicon 207 bp (1AL.1RS translocation) (Table 2).

The presence of *Yr17* was examined with the marker VENTRIUP/LN2 [42]. Positive wheat genotypes with *Yr17* amplified a 259-bp fragment (Appendix A). Three wheat cultivars (Markiz, Gomer and Nemchinovskaya 85) were postulated to have *Yr17*. These cultivars were resistant to the three *Pst* isolates avirulent to *Yr17* and were susceptible to two isolates virulent to *Yr17* (Table 1 and Table 2).

STS marker, csLV34, was used for the detection of the presence of *Yr18* [43]. This marker is located 0.4 cM distal to *Yr18* [44] and had two allelic variants. The 150- and 229-bp bands indicated the presence and absence of *Yr18*, respectively (Appendix A). Twenty-five genotypes (35%) amplified 150-bp fragments (Table 2).

A microsatellite marker, Xgwm6, was used for the identification of *Yr25* [32,45]. Cvs Strubes Dickkopf, Heines Peko, Carstens V and Heines VII were used as positive controls (Table 1). All positive controls and Russian cultivars produced amplicons with sizes near 150 bp (Appendix A). Another allele was not detected in the positive genotypes. The reason for this could be the absence of gene *Yr25* in the differential cultivars used or the low effectiveness of markers for the identification of *Yr25*. Also, no diagnosing ability of Xgwm6 was reported for molecular studies of an Indian wheat genotype [32].

Two microsatellite markers, Wmc313 and Wmc776, were used for the identification of *Yr60*. This gene confers moderate resistance at both the seedling and the adult plant stages [46]. A positive control for this gene was not included in our study. According to Rani et al. [32], Wmc313 amplifies 180- and 200-bp alleles in positive genotypes, and wmc776 amplifies three alleles of 150, 160 and 170 bp, respectively. During the testing of Russian wheat cultivars with marker Wmc313, we obtained either one amplicon with a size near 200 bp or nothing (Appendix A). The presence of an allele of 150 bp was revealed with marker wmc776 (Appendix A). It is, therefore, concluded that *Yr60* is absent in Russian genotypes. The donor of this gene is the Mexican wheat line Almop [32], and genotypes with *Yr60* have not been used in wheat breeding in Russia.

Many results (absence of certain *Yr* genes and presence of others) are consistent with the results reported for Western and Central Europe. However, the proportion of cultivars presenting some of the *Yr* is notably different.

## 3. Discussion

Over the last two decades, with the emergence of new *Pst* races, wheat yellow rust has been posing a serious threat to wheat production worldwide. To mitigate this threat, intensive global efforts are underway to carefully monitor the evolution and dispersal of *Pst* races, determine the pathotypes of isolates, screen disease-resistant germplasm and breed wheat cultivars with durable disease resistance [22].

In this study, 72 modern Russian commercial winter wheat cultivars were evaluated using *Pst* isolates. Multi-pathogen tests were combined with DNA marker data to postulate the presence of *Yr* genes. The disease reaction types of the cultivars differed significantly following inoculation with *Pst* isolates from geographically separated distant regions (North Caucasus, North-West and West Siberia); regions that are separated by thousands of kilometers. The number of winter wheat cultivars resistant to the West Siberian *Pst* isolate was significantly higher than to North Caucasian and North-West *Pst* isolates where disease outbreaks occur annually.

In the past five years, the significance of yellow rust has notably risen in all areas of the North Caucasus region. It is detected annually in spring from stem extension stages to heading, which results in the need to apply fungicidal treatments [47]. In the North-West region, disease appears in the last half of June at the flowering stage of winter wheat, and the maximum of the *Pst* development is detected at the ripening stage. A similar situation with yellow rust incidence has been observed in the region of Central Chernozem [21]. The disease is observed sporadically in the Western Siberia and Volga region on spring wheat [19,20].

Thirteen wheat cultivars had resistance to all *Pst* races tested at the seedling stage. According to the Russian State register [34,35,36,37] most of these are characterized as having field resistance. This indicates that these cultivars could contain (1) a single (or multiple) effective gene(s) to which all pathotypes are effective, (2) an effective combination of genes that have lost their effectiveness in their single form or (3) a very high durable resistance that may be a combination of race-specific genes inducing resistance at the seedling stage and/or genes with quantitative effects. When molecular markers were used to identify the highly effective resistance genes *Yr5*, *Yr10*, *Yr15* and *Yr24*, indicative DNA fragments were observed only in the positive controls, indicating that none of the entries had these genes. Virulence to these genes is not known in Russia [39]. Resistance genes *Yr5* and *Yr15* remain effective against the predominant *Pst* races worldwide [13,33,48,49]. Virulence to *Yr10* and *Yr24* occurs in some regional populations around the globe, but at a low frequency [13,50]. Despite *Yr5*, *Yr10*, *Yr15* and *Yr24* not being detected in Russian cultivars, these genes are commonly deployed in some other countries [31,32].

In molecular analysis, the frequent presence of *Yr9* and *Yr18* was detected (17 and 35% of cultivars, respectively). The donors of seedling resistance gene *Yr9* and partial resistance gene (or slow-rusting genes) *Yr18* were first used in wheat breeding in 1970. The resistance breakdown of *Yr9* has occurred since the late 1980s due to the widespread emergence of virulent *Pst* races worldwide. This has resulted in major epidemics of yellow rust that have challenged world wheat production [51,52]. *Yr9* is closely linked with *Sr31*, *Lr26* and *Pm8*, and it is located on the short arm of rye chromosome 1 (1RS). It has been widely used in wheat by means of wheat-rye translocation chromosomes [53]. The 1BL.1RS translocation in Russian winter wheat cultivars was derived from cv. Kavkaz. Currently, wheat cultivars worldwide with *Yr9* are mostly susceptible to yellow rust. Despite this, donors of this translocation are still widely used in modern wheat breeding. The presence of this translocation may relate more to its connotation with widespread adaptability and higher wheat yield [54]. Two Russian genotypes had 1AL.1RS translocations of 1RS origin from the same rye cv. Insave. The 1RS translocation is derived from rye cv. Imperial has the stem rust resistance gene *SrR* but no known leaf or yellow rust or powdery mildew resistance genes [53].

Seedling resistance genes are detected during both the seedling and adult plant stages and as such constitute an all-stage resistance phenotype. The majority of genes that confer race-specific resistance to rusts and other biotrophic fungi (R genes) remain effective for only a few years when deployed at a larger scale. APR is commonly detected at the post-seedling stage and often as field resistance. A large proportion of seedling resistance genes exhibit phenotypes of major effect and with varying infection types, whereas most of the APR genes are partial in effect with varying levels of disease severity. Locus with genes *Yr18*, *Lr34*, *Sr38*, *Pm38* and *Bdv1* is globally used as a component of durable rust resistance in breeding programs. Flag leaves of many wheat cultivars containing this locus in certain environments develop a necrotic leaf tip. This morphological marker is referred to as leaf tip necrosis (*Ltn1*) [55,56]. The additive effects of these slow-rusting genes form the basis of durable resistance to rusts in wheat cultivars worldwide [44,57]. This gene was found either alone or in combination with *Yr8* in the cultivars tested. In Russia specifically, *Yr18* has lost effectiveness, but it still enhances resistance in combination with other resistance genes. Also, the same has been shown for leaf rust resistance [58,59]. Genotypes with three or more ineffective leaf rust genes (e.g., *Lr1*, *Lr3*, *Lr10* and *Lr20*) in combination with *Lr34* have higher resistance in the field than those with only one or two of these genes.

Molecular marker detection indicated that only 4% of the cultivars tested had *Yr17.* As indicated, *Yr17* has long been demonstrated to be ineffective in Western Europe through many studies but remains present in the material despite the virulence of many current *Pst* pathotypes [22]. This gene originated from *T. ventricosum* and is tightly linked to leaf rust resistance gene *Lr37*, stem rust resistance gene *Sr38* and yellow rust resistance gene *Yr17* [5]. The *Yr17* resistance gene has been incorporated into wheat cultivars in Northern Europe since the mid-1970s. Wheat cultivars with *Yr17* were first grown in the UK, Denmark, France and Germany in 1980–1990 [60]. Virulence to *Yr17* was detected in 1995, following intensive use of this single resistance gene in widely grown cultivars [61]. Currently, *Yr17*-virulence is common in northwestern European *Pst* populations. The Russian *Pst* isolates studied in 2019–2021 [39] were mostly avirulent to *Yr17*. Two isolates virulent to *Yr17* from North-West and Derbent were used in this study.

The use of molecular markers in wheat breeding is a fast and efficient way to confirm and detect the presence of resistance genes. They have an edge over the standard phenotyping as these are not affected by plant growth stages or the environment [62]. In the present study, DNA markers were used to postulate the presence of *Yr2*, *Yr5*, *Yr7*, *Yr9*, *Yr10*, *Yr15*, *Yr17*, *Yr18*, *Yr24*, *Yr25* and *Yr60*. Mostly, these markers were highly effective for the identification of *Yr* genes. However, some of them gave inconclusive results, which could have been due to some lack of specificity of the markers (e.g., marker Wmc364 for *Yr2*, CFD77 for *Yr7* and Xgwm6 for *Yr25*). A similar outcome for these markers was reported for other studies [32].

## 4. Materials and Methods

### 4.1. Plant Materials

Seventy-two winter wheat cultivars included in the Russian Register of Breeding Achievement (National List) and approved for use in seven geographically distant Russian regions (North-West, Central, Central Chernozem, Middle Volga, North Caucasus, Ural and West Siberia) in 2019–2022 were assessed in this study (Table 2). Of these, the North Caucasian cultivars dominated (63%). Disease epidemics in Russia have been frequent and destructive, mostly in the North Caucasus. The field resistance of this material to a range of diseases has been previously evaluated in state regional nurseries. Generalized characteristics of new wheat cultivars are presented in an official publication of the Russian Register [34,35,36,37,63].

### 4.2. Seedling Tests

Three to five seeds of each genotype were planted in 10 cm diameter plastic pots in a disease-free area. Twelve- to fifteen-day-old plants were used for yellow rust resistance assessments [54]. Urediniospores of a single isolate were suspended in Novec 7100 (3M, St. Paul, MN, USA) in a glass tube and connected to the airbrush spray gun. This suspension was sprayed onto seedlings (2-leaf stage) of each cultivar. The inoculated plants were incubated in a dark dew chamber at 10 °C for 24 h and then transferred to a growth chamber (Environmental Test Chamber MLR-352H, Sanyo Electric Co., Ltd., Osaka, Japan) with 16:8 h L:D photoperiod at 16 and 10 °C, respectively.

Five isolates of *P. striiformis* were used to inoculate the wheat cultivars at the seedling stage. These isolates were selected during virulence studies of Russian *Pst* populations in 2020–2022. The virulence–avirulence profiles are shown in Table 1. Cvs Jupateco S and Avocet S were used as susceptible controls.

Seedling infection type was scored after 16–18 days on a five-point scale [64]: 0, no visible uredia or necrotic areas without sporulation; 1, necrotic and chlorotic areas with restricted sporulation; 2, moderate sporulation with necrosis and chlorosis; 3, sporulation with chlorosis; and 4, abundant sporulation without chlorosis. More visual necrosis or chlorosis than the average for scores 1 to 3 are indicated by an appending a semicolon. Infection types 0 to 2 were considered resistant and 3 to 4 susceptible.

### 4.3. DNA Extraction, PCR Amplification and Electrophoresis

Genomic DNA was extracted by the method of Dorokhov and Klocke [65]. The DNA stock solution was adjusted to a concentration of 100–150 ng/mL with nuclease-free sterile water as the working concentration for the polymerase chain reaction (PCR) and stored at −20 °C. PCRs were performed using a thermocycler (C1000, BioRad, Hercules, CA, USA) in 20 mL of a PCR mixture containing 100–150 ng of genomic DNA, 2 units of *Taq* DNA polymerase, 1X PCR buffer (10 mM Tris HCL), 2.5 mM of MgCl_2_, 100 mM of each dNTP and 10 mM of each primer. The recommended PCR protocol was used in amplifications. PCR products were separated on 1.5 to 3% agarose gels (depending on gene product size) and visualized under UV light using the digital gel imaging system (GelDocGo, BioRad, Hercules, CA, USA).

All cultivars were evaluated with molecular markers (Table 3) linked to 11 known genes, *Yr2*, *Yr5*, *Yr7*, *Yr9*, *Yr10*, *Yr15*, *Yr17*, *Yr18*, *Yr24*, *Yr25* and *Yr60*.

## 5. Conclusions

The present study provides information on the resistance of modern Russian winter wheat cultivars to yellow rust and contributing diversity for *Yr* genes. Multi-pathogen tests were used for the determination of the resistance phenotype and molecular markers for the resistance gene presence (*Yr2*, *Yr5*, *Yr7*, *Yr9*, *Yr10*, *Yr15*, *Yr17*, *Yr18*, *Yr24*, *Yr25* and *Yr60*). Thirteen genotypes that were highly resistant to all used *Pst* isolates and resistant in the field were detected (17%). The resistance genes *Yr5*, *Yr10*, *Yr15* and *Yr24* were effective against all Russian *Pst* races. However, they were not found in these resistant cultivars. Consequently, these cultivars might have either a novel resistance gene(s) or an effective combination of other resistance genes.

The genes, *Yr9*, *Yr17* and *Yr18*, and translocation 1AL.1RS, widely used in wheat cultivars worldwide were present in the Russian material. Although, they have lost their effectiveness when used alone, they can still be used together, or along with many other *Yr* genes, to enhance genetic diversity and the overall level and durability of yellow rust resistance.

The information obtained in this study will be important for managing yellow rust in wheat through the incorporation of the currently available resistant genotypes, genes and markers in breeding of new cultivars with resistance to yellow rust, and for the study and interpretation of possible changes in virulence of *Pst* population.

## Figures and Tables

**Table 1 plants-12-03471-t001:** Characteristic of infection types of the Avocet NILs and wheat differentials during inoculation with *Puccinia striiformis* isolates from different Russian regions.

Wheat Accession	*Yr* Gene	NW	NC_D	NC_Kr	NC_R	WS
*Yr1*/6*Avocet S	*Yr1*	0	3	3	3	3
*Yr5*/6*Avocet S	*Yr5*	0	0	0	0	0
*Yr6*/6*Avocet S	*Yr6*	3	3	3	3	3
*Yr7*/6*Avocet S	*Yr7*	2;	1–2;	2–3;	3	2;
*Yr8*/6*Avocet S	*Yr8*	2–3;	3	3	3	3
*Yr9*/6*Avocet S	*Yr9*	3	3	3	3	3
*Yr10*/6*Avocet S	*Yr10*	0	0	0	0	0
*Yr15*/6*Avocet S	*Yr15*	0	0	0	0	0
*Yr17*/6*Avocet S	*Yr17*	2–3	3	0–1;	1–2;	0
*Yr18*/6*Avocet S	*Yr18*	3	3	3	3	3
*Yr24*/6*Avocet S	*Yr24*	2;	0–1;	0–1;	0–1;	0;
*Yr26*/6*Avocet S	*Yr26*	2;	0–1;	0–1;	0–1;	0;
*YrSP*/6*Avocet S	*YrSp*	0	0	0	0	0
*Yr27*/6*Avocet S	*Yr27*	2–3;	3	0;	0–2;	3
Chinese 166	*Yr1*	0	3	3	3	3
Lee	*Yr7*, *Yr+*	3	3–4	3	3	3
Heines Kolben	*Yr6*, *Yr2*	3–4	3–4	3	3	3
Vilmorin 23	*Yr3*, *Yr+*	2–3	2–3	3	3	3
Moro	*Yr10*, *YrMor*	0;	0	0	0	0
Strubes Dickkopf	*YrSD*, *Yr25*, *Yr+*	2	2	2–3	3	3
Suwon 92/Omar	*YrSu*, *Yr+*	3–4	3	3–4	3	3
Hybrid 46	*Yr4*, *Yr+*	2	3	3–4	3	2;
Reichersberg 42	*Yr7*, *Yr+*	3	2–3	3–4	3	2;
Heines Peko	*Yr2*, *Yr6*, *Yr25*, *Yr+*	3–4	2;	3–4	3	3
Nord Desprez	*Yr3*, *YrND*, *Yr+*	0–1;	0	2;	2;	2;
Compair	*Yr8*, *Yr19*	3–4	0–2;	2;	2	3
Carstens V	*Yr32*, *Yr25*, *Yr+*	0–1;	2–3	3–4	3	2;
Spaldings Prolific	*YrSP*, *Yr+*	0	0	0	0	0
Heines VII	*Yr2*, *Yr25*, *Yr+*	2–3	2–3	2–3	3	2–3;
Jupateco S, Avocet S	susceptible check	3–4	3–4	3–4	3–4	3–4

Origin of *P. striiformis* isolates: NW, North-West; NC, North Caucasus: D, Derbent; Kr, Krasnodar; R, Rostov; WS, West Siberia. Infection type 0 (immune), no visible uredia; 1 (resistant), small uredia with necrosis; 2 (resistant to moderately resistant), small to medium sized uredia with green areas and surrounded by necrosis or chlorosis; 3 (moderately resistant/moderately susceptible), medium sized uredia with or without chlorosis; 4 (susceptible), large uredia without chlorosis. Symbols (;) is larch hypersensitive flecks [5].

**Table 2 plants-12-03471-t002:** Characteristic of winter common wheat included in the Russian Register of Breeding Achievement in 2019–2022 for yellow rust resistance.

Cultivar	Production Region ^1^	Field Resistance ^2^	*Puccinia striiformis* Infection Types					*Yr* Genes
			NW ^3^	NC_D	NC_Kr	NC_R	WS	
2019								
Arsenal	NC, LV	R	2;	3	3	3	3	*Yr18*
Bazal’t 2	LV	-	3–4	3	2+–3	3	3	*Yr18*
Bodryy	C	-	3	3	2+–3	3	0–2;	
Donmira	NC	-	3	3	3	3	3	
Etyud	NC, LV	R	2+–3	0–1	2+–3	2+–3	3−	*Yr18*
Felitsiya	C	-	2–3	3	3	3	3	*Yr18*
Gerda	NC	R	3	3	3	3	1–2;	
Iridas	NC	-	2–3	3	0;	0–1	0	*Yr9*
Kavalerka	NC	R	3	3	3	3	3	
Korona	NC	-	3	3	3	3	3	*Yr9*, *Yr18*
Markiz	NC	R	3–4	3	2	2	0–2;	*Yr17*
Shef	NC	-	2+–3	2+–3	2+–3	2+–3	0	*Yr18*
Stat’	LV	R	2–3	3	3	3	0	*Yr18*
STRG 8060 15	CCh	-	3	2–3	3	3	3	
Timiryazevka 150	CCh, NC, LV	R	2	1–2;	1–2;	2	2;	*Yr9*
Videya	NC	R	3	3	3	3	0	*Yr9*
2020								
Akapella	CCh, NC	MR	2;	0–1	2	2	0	*Yr18*
Akhmat	CC, NC	R	3	3	2+–3	3	0	*Yr9*
Al’ternativa	MV	-	3	3	2–3	3	3	
Anastasiya	LV	-	3	2	2	2	1–2;	
Armada	CCh, NC	R	3	3	3	3	2–3	*Yr9*
Barynya	NC	-	3	2+–3	3–4	3	3	
Bylina Dona	NC	-	3	3	1–2;	2	3−	
Donskaya step’	NC, LV	-	3	3	2+–3	3	3	
Gomer	CCh, NC	YR	2;–3	3	0–1;	0–1	0;	*Yr17*
Paritet	LV	-	3	3	3	3	2+–3	*Yr9*, *Yr18*
Sekletiya	NC, LV	-	3	2+–3	3	3	3–4	*Yr18*
Tsefey	CCh	-	3	2+–3	2+–3	3	3	
Vol’nitsa	NC	MR	0	2-	0;–1	0–1	0–1;	*Yr18*
Vol’nyy Don	NC	-	3	3	3–4	3	3	*Yr18*
V’yuga	MV	-	2+–3	0	1–2;	2	2+–3	*Yr18*
Yelanchik	NC, LV	R	2	2	1–2	2	0–1;	
Yelanskaya	LV	-	3	3	3	3	3	
Zhavoronok	NC, LV	-	3	3	3	3	3	*Yr18*
2021								
Al’bireo	CCh	-	2;	2+–3	3	2+–3	3–4	*Yr18*
Bogema	NC	-	2+–3	2+–3	3	2+–3	1;–2	
Bumba	NC	MS	2;	3	2+–3	3	0;	
Galateya	C	-	0;	2	2	2;	0–1	
Khamdan	NC, LV	YR	2	2	2	2	2;	*Yr18*
Klassika	CCh, NC, LV	R	3	3	3	2;	3–4	
Krasnoobskaya ozimaya	WS	R	3	3	3	3	3–4	
Moskovskaya 82	VV, CCh	-	3	3	2	2;	0	
Nemchinovskaya 85	C, VV, CCh	-	3	3	0;	0–1;	0;	*Yr17*
Partner	NC	-	2+–3	3	3	3	2	*Yr18*
Polina	NC	MR	0–1;	2;	2;	1–2;	0–2;	*Yr9*
Rifey	U	-	2+–3	2+–3	2+–3	2+–3	1–2;	*Yr18*
Rossyp’	NC, LV	MS	2+–3	3	2+–3	2+–3	1–2;	*Yr9*
Sharm	NC	R	2;	2	2;	2+;	2	
Status	NC, LV	-	2;	2;	2+–3;	2+–3	0	*Yr9*, *Yr18*
Stil’ 18	CCh, NC, LV	R	2+–3	3	2+–3	3–4	2+–3	
Taygeta	CCh, MV	-	2;	3	2;	2	2	*Yr18*
Yubiley Dona	NC, LV	MR	2;	2	2	1–2;	2	
2022								
Agrofak 100	CCh, NC	R	3–4	3	3	3	2+–3	
Ambar	CCh, NC	-	3	3	3	3	3–4	*Yr9*
Batya	CCh, NC, LV	-	3–4	3	3	3–4	3	
En Foton	CCh	-	3–4	2	0	0–1	2	*Yr18*
En Mars	CCh	-	3–4	1–2;	3	3	3–4	
Estafeta	MV	-	3–4	2+–3	3-	2+–3	2+–3	
Fyodor	CCh, NC	R: YR	2	0	0–1;	2	2;	*Yr9*
Leo	NC	-	1–2;	2+–3	2+–3	2+–3	2+–3	*Yr9*
Mig	CCh, NC	MR:YR	3–4	3	3	3	3	
Morets	NC, LV	R	3	3	2+–3	2+–3	3	*Yr18*
Pal’mira 18	CCh, NC, MV	MR	2	0	0	0–1;	2	*Yr18*
Podruga	LV, U	-	2;	0	2;	2;	2;	*Yr18*
Shkola	CCh, NC, MV, LV	MR	3	3	2–3	3	2–3	*1AL.1RS*
Sirena	CCh	-	2;	0	0–1;	2	0–1;	
Studencheskaya niva	LV	-	3–4	3	3	2+–3	2+–3	*Yr18*
Timiryazevskaya yubileynaya	CCh	-	3–4	3	0–1;	1–2;	3–4	
Vladi	C, VV	-	3–4	3	3	3–4	2–3	
Volodya	CCh, NC, MV	-	3–4	3	0	0–2;	3	
Zarechnaya	CCh	-	3–4	3	3	3–4	2–3	*1AL.1RS*
Zodiak	NC	-	3	2+–3	3	3	3	

^1^ Regions of the Russian Federation: NW, North-West; C, Central; CCh, Central Chernozem; MV, Middle Volga; NC, North Caucasus; U, Ural; WS, West Siberia. ^2^ According to the characteristics presented in the official publications “Characteristics of plant varieties included in Public Register of Breeding Achievements” [34,35,36,37] and the catalogue of the National Center of Grain named P.P. Lukyanenko [38]. R—resistant; MR—moderately resistant; MS—moderately susceptible; - no information in the Russian Register of Breeding Achievement. ^3^ NW, North-Western *P. striiformis* isolate; NC_D, North Caucasian from Derbent; NC_Kr, North Caucasian from Krasnodar; NC_R, North Caucasian from Rostov; WS, West Siberian. Symbols (+) and (−) indicate slightly larger and smaller pustule sizes, respectively [5].

**Table 3 plants-12-03471-t003:** Molecular markers used for identification of *Yr* genes.

Gene	Marker	Primer Sequence	References
*Yr2*	Wmc364	ATCACAATGCTGGCCCTAAAAC CAGTGCCAAAATGTCGAAAGTC	[40]
*Yr5*	STS7/8	GTACAATTCACCTAGAGT GCAAGTTTTCTCCCTATT	[66]
	STS9/10	AAAGAATACTTTAATGAA CAAACTTATCAGGATTAC	
*Yr7*	CFD77	CTGCTTCAGGGATTGGAGAG GTTTCCTGGGCTAAACCACA	[32]
*Yr9*	SCM9	TGAСААСССССТТТСССТCGT ТСАTCGACGСТАAGGAGGАССС	[41]
*Yr10*	Xpsp3000	GCAGACCTGTGTCATTGGTC GATATAGTGGCAGCAGGATACG	[67]
*Yr15*	Xbarc8	GCGGGAATCATGCATAGGAAAACAGAA GCGGGGGCGAAACATACACATAAAAACA	[68]
*Yr17*	Ventriup LN2	AGGGGCTACTGACCAAGGCT TGCAGCTACAGCAGTATGTACACAAAA	[42]
*Yr18*	csLV34	GTTGGTTAAGACTGGTGATGG TGCTTGCTATTGCTGAATAGT	[43]
*Yr24*	Barc181	CGCTGGAGGGGGTAAGTCATCAC CGCAAATCAAGAACACGGGAGAAAGAA	[32,69]
*Yr25*	Xgwm6	CGTATCACCTCCTAGCTAAACTAG AGCCTTATCATGACCCTACCTT	[32,45]
*Yr60*	Wmc776	CCATGACGTGACAACGCAG ATTGCAGGCGCGTTGGTA	[46]
	Wmc313	GCAGTCTAATTATCTGCTGGCG GGGTCCTTGTCTACTCATGTCT	

## Data Availability

All data are provided in the manuscript.

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
