# Peer review of "Resistance of Modern Russian Winter Wheat Cultivars to Yellow Rust"

_plants, 2023, doi:10.3390/plants12193471_

Round 1
Reviewer 1 Report
Comments for authors
Abstract
L9: method
L10: may be “included”?
L15: all Pst isolates… I suggest adding the term "tested”, especially if other isolates are observed in Russia
L18: may be rather “these genes are no longer effective now when deployed…”
Introduction
L30: may be “areas” rather than “zones”
L41: climate adaptability
L44-51: These sentences are about other rust species, I suggest focusing on stripe rust on the Introduction
L55-57: several publications partly explain this phenomenon, including:
Mboup, M., Bahri, B., Leconte, M., De Vallavieille‐Pope, C., Kaltz, O., & Enjalbert, J. (2012). Genetic structure and local adaptation of European wheat yellow rust populations: the role of temperature‐specific adaptation. Evolutionary applications, 5(4), 341-352.
L58-60: you cite numerous publications to justify this sentence, many of which do not seem specifically relevant to justify it, relating more to yellow rust in general. I suggest you reduce this number, and focus on publications that justify your point, an example:
de Vallavieille-Pope, C., Ali, S., Leconte, M., Enjalbert, J., Delos, M., & Rouzet, J. (2012). Virulence dynamics and regional structuring of Puccinia striiformis f. sp. tritici in France between 1984 and 2009. Plant Disease, 96(1), 131-140.
L60-63: same comment, some publications are not the most relevant; please reduce if possible the number of cited publications.
L60-63: to justify the end of the sentence, see for example:
Perronne, R., Dubs, F., de Vallavieille-Pope, C., Leconte, M., du Cheyron, P., Cadot, V., ... & Enjalbert, J. (2021). Spatiotemporal changes in varietal resistance to wheat yellow rust in France reveal an increase in field resistance level during the period 1985–2018. Phytopathology, 111(9), 1602-1612.
L72-73: again, I suggest considering other publications concerning race-specific resistance gene to justify the duration of resistance of these genes, for instance, as previously suggested above:
de Vallavieille-Pope, C., Ali, S., Leconte, M., Enjalbert, J., Delos, M., & Rouzet, J. (2012). Virulence dynamics and regional structuring of Puccinia striiformis f. sp. tritici in France between 1984 and 2009. Plant Disease, 96(1), 131-140.
L95-96: this information is given above in the paragraph
General comment: please check the relevance of these terms in their context in the Introduction, and rephrase: resilient cereal…, the leader among…, genetic protection…, plastic and distributive rust species, alteration of the disease incidence and severity…, distribution of Yr genes
General comment: given the two methods used to characterize the resistance of wheat varieties to yellow rust, I suggest to develop these points in the Introduction so that the reader understands the complementarity of these two approaches. Furthermore, it seems important to further develop knowledge on the resistance of wheat varieties and races of yellow rust in Russia in order to better understand the question considered. Do we already have information on common resistances, breakdown resistances, current and past races of yellow rust and their virulence spectra in Russia?
Material and methods
Table 1: I strongly suggest removing the information about the resistance level concerning other pathogens from the table (column “Field resistance”) to avoid any false interpretation.
Table 1: In the "Yr genes" column, please specify the method used to obtain the information in the table caption so that it can be read independently of the Material and methods section.
Table 2: “supplemental” is not clear, please modify
Table 2: I suggest to better explain this table in the caption. For any non-pathologist or non-breeder reader, this will be difficult to understand independently of the Material and methods section. Furthermore, the more general interpretation is only given in the M&M section.
Results
General comment: Why did you start by presenting the results in Table 1 rather than Table 2? The description of the characteristics of the five isolates would be more relevant before that of the responses of the cultivars.
L121-123: repeating my previous comment, I can only suggest focusing on the yellow rust information only in the "field resistance" column.
L168: “characterized” rather than “had”
L172: same comment
L214-216: AvYr5 or AvYr7, please clarify
General comments: The search for race-specific genes appears interesting; I would like to make various comments and suggestions:
- Part of the current development of Results concerning markers should be repositioned in the M&M section
- Many results (absence of certain Yr, presence of others Yr) are very consistent with the results in Western and Central Europe
- But the proportion of varieties presenting some of the Yr is quite different
L253: remove “the” in “the absent in Russia”
Discussion
L264-267: please rephrase the sentence to avoid repetition
L267-269: this information could be given more clearly in the Results section
L279: all Pst races tested at the seedling stage. Adding the term "tested" is necessary, otherwise is probable that all varieties would be resistant according to the register.
L280-…: the structure of the paragraphs could be different. The fact that wheat varieties are highly resistant in the field can be explained by several possibilities, among which:
- one (or more) effective gene for which all pathotypes are avirulent,
- a combination of effective genes for which each pathotype does not have all of the virulence genes,
- a very high durable resistance which may be a combination of race-specific genes inducing resistance at the seedling stage and/or genes with quantitative effects
In a recent publication cited above, it has been shown that (1) despite repeated breakdown of race-specific Yr resistance genes, field resistance was only slightly affected in France over the last decades, and that would be due to the accumulation of both quantitative resistance and different race-specific resistance genes (Perronne, R., Dubs, F., de Vallavieille-Pope, C., Leconte, M., du Cheyron, P., Cadot, V., ... & Enjalbert, J. (2021). Spatiotemporal changes in varietal resistance to wheat yellow rust in France reveal an increase in field resistance level during the period 1985–2018. Phytopathology, 111(9), 1602-1612)
L325-334: the case of Yr17 is really interesting and I suggest developing this point further. As indicated, Yr17 has been breakdown for a long time in Western Europe, quite widely studied, and is currently very present in the material despite the virulence of many current Pst pathotypes.
Conclusions
Author Response
Dear reviewer,
Thank you for your positive assessment of our work. We agree with the most of notes. The answers to the comments are presented below.
With kind regards,
Elena Gultyaeva
Answers to comments.
L9: method Corrected
L10: may be “included”? Corrected
L15: all Pst isolates… I suggest adding the term "tested”, especially if other isolates are observed in Russia Corrected
L18: may be rather “these genes are no longer effective now when deployed…”
It was changed “While these genes have lost effectiveness individually, they can still enhance genetic diversity and overall yellow rust resistance, whether used in combination with each other or alongside other Yr genes.”
Introduction
L30: may be “areas” rather than “zones” Corrected
L41: climate adaptability Corrected
L44-51: These sentences are about other rust species, I suggest focusing on stripe rust on the Introduction. We have shortened this paragraph.
L55-57: several publications partly explain this phenomenon, including: Mboup, M., Bahri, B., Leconte, M., De Vallavieille‐Pope, C., Kaltz, O., & Enjalbert, J. (2012). Genetic structure and local adaptation of European wheat yellow rust populations: the role of temperature‐specific adaptation. Evolutionary applications, 5(4), 341-352. This publication was added.
L58-60: you cite numerous publications to justify this sentence, many of which do not seem specifically relevant to justify it, relating more to yellow rust in general. I suggest you reduce this number, and focus on publications that justify your point, an example: de Vallavieille-Pope, C., Ali, S., Leconte, M., Enjalbert, J., Delos, M., & Rouzet, J. (2012). Virulence dynamics and regional structuring of Puccinia striiformis f. sp. tritici in France between 1984 and 2009. Plant Disease, 96(1), 131-140.
We disagree with this remark. The proposed publication provides information about the changes in Pst population only in France. We present a more extensive geography of the Pst epidemic. We have reduced the number of references, but not so much.
L60-63: same comment, some publications are not the most relevant; please reduce if possible the number of cited publications.
L60-63: to justify the end of the sentence, see for example: Corrected
Perronne, R., Dubs, F., de Vallavieille-Pope, C., Leconte, M., du Cheyron, P., Cadot, V., ... & Enjalbert, J. (2021). Spatiotemporal changes in varietal resistance to wheat yellow rust in France reveal an increase in field resistance level during the period 1985–2018. Phytopathology, 111(9), 1602-1612. Corrected.
L72-73: again, I suggest considering other publications concerning race-specific resistance gene to justify the duration of resistance of these genes, for instance, as previously suggested above:
de Vallavieille-Pope, C., Ali, S., Leconte, M., Enjalbert, J., Delos, M., & Rouzet, J. (2012). Virulence dynamics and regional structuring of Puccinia striiformis f. sp. tritici in France between 1984 and 2009. Plant Disease, 96(1), 131-140. Corrected.
L95-96: this information is given above in the paragraph It was deleted.
General comment: please check the relevance of these terms in their context in the Introduction, and rephrase: resilient cereal…, the leader among…, genetic protection…, plastic and distributive rust species, alteration of the disease incidence and severity…, distribution of Yr genes. It was corrected
General comment: given the two methods used to characterize the resistance of wheat varieties to yellow rust, I suggest to develop these points in the Introduction so that the reader understands the complementarity of these two approaches. Furthermore, it seems important to further develop knowledge on the resistance of wheat varieties and races of yellow rust in Russia in order to better understand the question considered. Do we already have information on common resistances, breakdown resistances, current and past races of yellow rust and their virulence spectra in Russia? This information was added.
Material and methods
Table 1: I strongly suggest removing the information about the resistance level concerning other pathogens from the table (column “Field resistance”) to avoid any false interpretation. Corrected
Table 1: In the "Yr genes" column, please specify the method used to obtain the information in the table caption so that it can be read independently of the Material and methods section. The information about the resistance level concerning other pathogens was deleted from the table 2.
Table 2: “supplemental” is not clear, please modify Corrected
Table 2: I suggest to better explain this table in the caption. For any non-pathologist or non-breeder reader, this will be difficult to understand independently of the Material and methods section. Furthermore, the more general interpretation is only given in the M&M section. Corrected
Results
General comment: Why did you start by presenting the results in Table 1 rather than Table 2? The description of the characteristics of the five isolates would be more relevant before that of the responses of the cultivars. It was changed.
L121-123: repeating my previous comment, I can only suggest focusing on the yellow rust information only in the "field resistance" column. Corrected
L168: “characterized” rather than “had” Corrected
L172: same comment Corrected
L214-216: AvYr5 or AvYr7, please clarify. See the sentence above “According to McIntosh et al. [5], Yr7 is allelic or closely linked with Yr5.” It is correct.
General comments: The search for race-specific genes appears interesting; I would like to make various comments and suggestions:
- Part of the current development of Results concerning markers should be repositioned in the M&M section We disagree with this note
- Many results (absence of certain Yr, presence of others Yr) are very consistent with the results in Western and Central Europe
- But the proportion of varieties presenting some of the Yr is quite different We added these sentences in Results
L253: remove “the” in “the absent in Russia” Corrected
Discussion
L264-267: please rephrase the sentence to avoid repetition Corrected
L267-269: this information could be given more clearly in the Results section Corrected
L279: all Pst races tested at the seedling stage. Adding the term "tested" is necessary, otherwise is probable that all varieties would be resistant according to the register. Corrected
L280-…: the structure of the paragraphs could be different. The fact that wheat varieties are highly resistant in the field can be explained by several possibilities, among which:
- one (or more) effective gene for which all pathotypes are avirulent,
- a combination of effective genes for which each pathotype does not have all of the virulence genes,
- a very high durable resistance which may be a combination of race-specific genes inducing resistance at the seedling stage and/or genes with quantitative effects We added these sentences.
In a recent publication cited above, it has been shown that (1) despite repeated breakdown of race-specific Yr resistance genes, field resistance was only slightly affected in France over the last decades, and that would be due to the accumulation of both quantitative resistance and different race-specific resistance genes (Perronne, R., Dubs, F., de Vallavieille-Pope, C., Leconte, M., du Cheyron, P., Cadot, V., ... & Enjalbert, J. (2021). Spatiotemporal changes in varietal resistance to wheat yellow rust in France reveal an increase in field resistance level during the period 1985–2018. Phytopathology, 111(9), 1602-1612)/
It was corrected. Information was added.
L325-334: the case of Yr17 is really interesting and I suggest developing this point further. As indicated, Yr17 has been breakdown for a long time in Western Europe, quite widely studied, and is currently very present in the material despite the virulence of many current Pst pathotypes.
Corrected.

Reviewer 2 Report
This study aimed to assess the yellow rust resistance potential of modern common winter wheat cultivars. To achieve this, seedling tests were conducted using various Puccinia striiformis races and molecular markers associated with Yr resistance genes. The work was carried out at a high methodological level, and all conclusions were substantiated. The undoubted novelty of the work is the testing of new wheat cultivars for resistance to yellow rust from several regions of the Russian Federation. However, I have a request to the authors to recheck all the values in the text - the number of isolates, cultivars, genotypes, and primer sequences, as well as to be more attentive to the correct spelling of geographical names. For clarity, see the text of the manuscript.

Author Response
Dear reviewer,
Thank you for your positive assessment of our work. We agree with the most of notes.
We accepted all editorial-type changes.
A revised version of our manuscript was prepared accordingly.
Thank you very much for the comments and suggestions to improve our manuscript.
We would like to answer for two notes below:
- “Identification of genes Yr5, Yr10, Yr15 and Yr24 was done for twelve resistant cultivars….” Why didn't you search for YrSp and Yr26? They have also demonstrated their effectiveness.
According to the literature data, the Yr24 and Yr26 genes are identical. The reaction of isogenic lines to Yr24 and Yr26 has always been similar.
- Why do not you try to find these genes in other studied cultivars? Because these gene are highly effective. No virulence isolates were found in the Russian Pst populations. Other cultivars were susceptible from one to all of the used Pst isolates used, which determined the absence of these genes in these genotypes.
With kind regards,
Elena Gultyaeva, Ekaterina Shaydayuk
Round 2
Reviewer 1 Report
Comments for authors
General comment: please check the white spaces in the sentences in the new version of the manuscript
Abstract
L16: I suggest to remove “in resistant cultivars”, not useful in the sentence
Introduction
General comment: Most comments suggest rewritings of sentences, often unclear. The general structure of the Introduction is appropriate.
L25: is the most cultivated Triticum…
L59: warmer temperatures influenced the yellow rust resistance…
L71: remove ]
L73: usually characterized by…
L79: with these genes mostly…
L90: and characterized by genes…
L95: a literature reference would be welcome. Below, I cite two works of my knowledge (modeling and analysis of the sources of resistance of the commercial variety “Apache”), but similar work may have been carried out by your institute.
Elisabeth Lof, M., de Vallavieille-Pope, C., & van Der Werf, W. (2017). Achieving durable resistance against plant diseases: scenario analyses with a national-scale spatially explicit model for a wind-dispersed plant pathogen. Phytopathology, 107(5), 580-589.
Paillard, S., Trotoux-Verplancke, G., Perretant, M. R., Mohamadi, F., Leconte, M., Coedel, S., ... & Dedryver, F. (2012). Durable resistance to stripe rust is due to three specific resistance genes in French bread wheat cultivar Apache. Theoretical and Applied Genetics, 125(5), 955-965.
L97: the cited reference [22] also includes seedling tests, I suggest removing this quote here to make the sentence more coherent
L98: the field resistance level of Russian wheat genotypes concerning the most important diseases is also evaluated…
L101-102: the sentence is not clear, please rephrase
L107: commercial
L108: is too limited…
Material and methods
L361: seven
Results
Table 1: Characteristics of infection of the Avocet NILs and wheat differential during inoculation…
L117: Table 2 or Table 1? Please verify
L174: in comparison with…
L195: not sure about the use of the term “supplemental” in this context
L224: Infection types of Pst isolates used in this study…
L226-227: the sentence is not clear, please rephrase
L267: some of the Yr genes…
Discussion
L273: I suggest to citer other complementary references previously cited in your manuscript, such as, for examples:
Cowger, C., & Brown, J. K. (2019). Durability of quantitative resistance in crops: greater than we know?. Annual review of phytopathology, 57, 253-277.
L280: change italicized word
L291: these cultivars could be characterized by…
L291-292: the points (1) and (2) are not clear ; not sure if you want to say “(1) a single (or multiple) Yr gene(s) effective against all Pst pathotypes studied, (2) a combination of Yr genes for which none Pst pathotype studied has all of the avirulence genes”, rephrase if necessary
Conclusions
Author Response
Dear reviewer,
Thank you for new corrections of our manuscript. We agree with the most of notes and corrected all of them.
A revised version of our manuscript was prepared accordingly.
With kind regards,
Elena Gultyaeva Ekaterina Shaydayuk